

# Prevalence and characterization of extended-spectrum β-lactamase-producing *Escherichia coli* and *Klebsiella pneumoniae* isolated from raw vegetables retailed in Southern Thailand

Chonticha Romyasamit[1,*], Phoomjai Sornsenee[2,*], Siriphorn Chimplee[3], Sitanun Yuwalaksanakun[1], Dechawat Wongprot[1] and Phanvasri Saengsuwan[3]

[1] Department of Medical Technology, School of Allied Health Sciences, Walailak University, Nakhon Si Thammarat, Thailand

[2] Department of Family and Preventive Medicine, Faculty of Medicine, Prince of Songkla University, Songkhla, Thailand

[3] Department of Biomedical Sciences and Biomedical Engineering, Faculty of Medicine, Prince of Songkla University, Songkhla, Thailand

[*] These authors contributed equally to this work.

Corresponding author
Chonticha Romyasamit,
chonticha.ro@wu.ac.th

## ABSTRACT

**Background.** The increasing prevalence of broad-spectrum ampicillin-resistant and third-generation cephalosporin-resistant *Enterobacteriaceae*, particularly *Escherichia coli* and *Klebsiella pneumoniae*, has become a global concern, with its clinical impacts on both human and veterinary medicine. This study examined the prevalence, antimicrobial susceptibility, and molecular genetic features of extended-spectrum β-lactamase (ESBL)-producing *E. coli* and *K. pneumoniae* isolates from 10 types of raw vegetables.

**Methods.** In total, 305 samples were collected from 9 markets in Nakhon Si Thammarat, Thailand, in 2020.

**Results.** ESBL-producing *E. coli* and *K. pneumoniae* isolates were found in 14 of the 305 samples obtained from 7 out of 10 types of vegetables (4.6% of the total). Further, 14 ESBL-producing *E. coli* ($n = 5/14$) and *K. pneumoniae* isolates ($n = 9/14$) (1.6% and 3.0%, respectively) were highly sensitive to β-lactam/carbapenem antibiotics (imipenem, 100%). ESBL-producing *E. coli* ($n = 4$) and *K. pneumoniae* isolates ($n = 8$) were also sensitive to non-β-lactam aminoglycosides (amikacin, 80.00% and 88.89%, respectively). ESBL producers were most resistant to β-lactam antibiotics, including ampicillin (85.71%) and the cephalosporins cefotaxime and ceftazidime (64.29%). The most frequently detected gene in ESBL-producing *E. coli* and *K. pneumoniae* was $bla_{SHV}$. However, two ESBL-producing *E. coli* isolates also carried three other ESBL-encoding variants, $bla_{TEM}$, $bla_{CTX-M1}$, $bla_{GES}$ and $bla_{TEM}$, $bla_{SHV}$, $bla_{CTX-M9}$, which may be due to their association with food chains and humans.

**Discussion.** Indeed, our results suggest that raw vegetables are an important source of ESBL-resistant *E. coli* and *K. pneumoniae*, which are potentially transmittable to humans via raw vegetable intake.

## INTRODUCTION

Extended-spectrum β-lactamase (ESBL)-producing *Enterobacteriaceae*, particularly *Escherichia coli* and *Klebsiella pneumoniae*, produce nosocomial infections in patients and can also affect communities of heathy individuals (*Nji et al., 2021*; *Falagas & Karageorgopoulos, 2009*; *Boonyasiri et al., 2014*). The increasing prevalence of ESBL-producing strains has increased the possibility of the development of β-lactam antibiotic resistance (*Falagas & Karageorgopoulos, 2009*; *Pitout & Laupl, 2008*). This will exacerbate global public health problems and thus requires resolution. Apart from nosocomial infections, ESBL-producing *E. coli* and *K. pneumoniae* have emerged as community infection concerns, causing severe infections, such as urinary and respiratory tract infections and bacteremia (*Falagas & Karageorgopoulos, 2009*; *Pitout & Laupl, 2008*; *Teklu et al., 2019*). Therefore, it is crucial to identify the sources of ESBL-producing *E. coli* and *K. pneumoniae*. In the last decade, the prevalence of resistance to aminoglycosides, sulfonamides, fluoroquinolones, and cephalosporins in ESBL-producing *E. coli* and *K. pneumoniae* has increased considerably, limiting the treatment of infections caused by these bacteria (*Chong, Shimoda & Shimono, 2018*; *Wu et al., 2013*). Currently, more than 350 variants of ESBL have been identified using amino acid sequence comparisons and are categorized into 9 distinct families (*Yazdi et al., 2012*). The four most common ESBLs, namely $bla_{CTX-M}$, $bla_{TEM}$, $bla_{SHV}$, and $OXA$, are found in *Enterobacteriaceae* species (*Guenther, Ewers & Wieler, 2011*). $bla_{TEM}$ is predominant in China, whereas $bla_{SHV}$ is the predominant variant of ESBLs in Canada. Moreover, $bla_{CTX-M}$ ispredominant in various countries such as Spain, the United States of America, and the United Kingdom (*Yazdi et al., 2012*). In a previous study, three $bla_{CTX-M-15}$ variants, one $bla_{CTX-M-1}$ variant, two $bla_{CTX-M-9}$ variants, and one $bla_{SHV}$ variant were found in retail raw vegetables (*Reul et al., 2014*).

Raw vegetables are considered exceptional human food due to their convenience for uncooked consumption (*Brookie, Best & Conner, 2018*). In addition, consumers deem vegetables more advantageous to health; thus, there has been a noted increase in the consumption of vegetables instead of foods produced from animals (*Brookie, Best & Conner, 2018*). However, uncooked vegetables can have high levels of microbial contamination, which in turn could lead to a high rate of cross-contamination events (*E. coli* and *K. pneumoniae*–vegetables–human) (*Boonyasiri et al., 2014*; *Kim et al., 2015*; *Mritunjay & Kumar, 2017*; *Saksena, Malik & Gaind, 2020*). In fact, it has been reported that ESBL-producing bacteria could be present in fresh vegetables such as iceberg lettuce, spinach, and tomato (*Blaak et al., 2014*; *Richter et al., 2019*). However, studies on the incidence of ESL-producing *Enterobacteriaceae* in fresh vegetables, which are commonly found and consumed in tropical countries such as Thailand, remain to be limited.

Previously, numerous studies have reported the prevalence of ESBL-producing *Enterobacteriaceae* bacteria; in particular, *E. coli* and *K. pneumoniae* have been found in human and animal foods and in food-producing animals (*Boonyasiri et al., 2014*; *Eibach et al., 2018*; *Ryu et al., 2012*; *Le et al., 2015*; *Abayneh et al., 2019*; *Montso et al., 2019*; *Ye et al., 2018*; *Tekiner & Özp1nar, 2016*; *Hiroi et al., 2012*; *Mesa et al., 2006*; *Kaesbohrer et al., 2019*). The isolation of both ESBL-producing species from vegetable samples has also been described in various studies (*Boonyasiri et al., 2014*; *Kim et al., 2015*; *Blaak et al., 2014*; *Richter et al., 2019*; *Ye et al., 2018*; *Kaesbohrer et al., 2019*; *Bhutani et al., 2015*; *Zurfluh et al., 2015*; *Song et al., 2020*). Although the prevalence and role of ESBL producers in vegetables, such as cabbages, cucumbers, tomatoes, coriander, and lettuce, which are cultivated ubiquitously worldwide, have been examined, there have been few studies on vegetables (yardlong beans, winged beans, basil, eggplant, and young cashew leaves) that are specifically found in Thailand and tropical countries, and are frequently consumed in Southern Thailand. Thus, this study aimed to examine the prevalence and characteristics of ESBL-producing *E. coli* and *K. pneumoniae* in common raw vegetables found in Southern Thailand. The associations between ESBL-encoding genes and ESBL-producing strains were characterized to correlate data with those previously obtained in isolates from other food and clinical sources.

# MATERIALS & METHODS

## Vegetable collection

In total, 305 samples derived from 10 common edible vegetables were included in this analysis: Thai yardlong beans (34), Thai eggplant (44), winged bean (25), young cashew leaves (20), Thai basil (36), cabbage (21), cucumber (36), tomato (31), long coriander (32), and lettuce (26). These were bought randomly between September 2020 and October 2020 from four and five local retail markets in Tha Sala and Mueang districts, respectively, in Nakhon Si Thammarat, Thailand. After purchase, the vegetable samples were collected in sterile containers, maintained at 4 °C, and tested within 24 h.

## Isolation and Identification of ESBL-producing *E. coli* and *K. pneumoniae*

About 25 g of each sample was weighed, suspended in 225 mL of 0.1% buffered peptone water (Oxoid, Hampshire, UK), and incubated at 37 °C for 24 h. These microbial enrichments were streaked onto ESBL-selective agar (HiMedia Laboratories, Mumbai, India), followed by incubation at 37 °C for 24 h under aerobic conditions to enable morphological and color-coded distinction of individual ESBL-producing *Enterobacteriaceae* isolates according to the specifications of the producer. Suspected colonies of the *Enterobacteriaceae E. coli* (pink to purple) and *K. pneumoniae* (bluish green) were picked from each selective plate and streaked onto nutrient agar (Oxoid) at 37 °C for 24 h, followed by biochemical identification using Vitek2 (bioMérieux, Marcy 1′ Etoile, France). All the *E. coli* and *K. pneumoniae* isolates were then identified and confirmed by matrix-assisted laser desorption ionization/time-of-flight mass spectrometry (MADLI-TOF

MS) (*Bubpamala et al., 2018*). All the confirmed isolates were stored at 2 °C–8 °C until antimicrobial drug susceptibility testing and evaluation of ESBL production.

## Screening of presumptive ESBL-producing isolates

The standard disk diffusion method was performed by placing β-lactam ring drugs ceftazidime (CAZ) (30 μg) and cefotaxime (CTX) (30 μg) on disks. The amount of bacteria was adjusted by 0.5 McFarland standard, and the suspension was inoculated onto Mueller-Hinton agar (MHA; Oxoid) using a sterile cotton swab. Thereafter, the drug disks were placed on inoculated plates, followed by incubation at 37 °C for 24 h. Isolates presenting an inhibition zone to CTX ($\leq$27 mm) and CAZ ($\leq$ 22 mm) around the disks were regarded as presumptive ESBL-producing strains, based on Clinical and Laboratory Standards Institute (CLSI) 2019 guidelines (*CLSI, 2019*).

## Confirmation of ESBL-producing isolates

The double-disk synergy method was performed to confirm ESBL production in the presumptive positive isolates. *E. coli* and *K. pneumoniae* suspensions were placed onto MHA. Then, 30-μg CTX and CAZ disks were placed in the center of the plate, followed by placing CTX and CAZ plus clavulanic acid (30/10 μg) disks at a distance of 20 mm from the central disk in the same plate (CLSI 2019). All plates were incubated at 37 °C for 24 h. Isolates were considered ESBL producers when the zone of inhibition with CTX or CAZ disk with clavulanic acid was increased by $\geq$5 mm in comparison to that by CTX or CAZ alone.

## Antibiotic sensitivity test

To assess the antibiotic sensitivity profile of ESBL-producing *E. coli* and *K. pneumoniae* were isolated from the vegetables, antibiotic sensitivity test was conducted according to the methods in the CLSI 2019 guidelines (*CLSI, 2019*) and *Abayneh et al. (2019)*, with modifications isolates were suspended and inoculated onto MHA, and then β-lactam ring antibiotic disks were placed on the culture plates. These plates contained 10 μg each of ampicillin (AMP), imipenem (IPM), meropenem (MEM), tetracycline (TE), gentamicin (CN), or cefpodoxime (CPD) or 30 μg each of amikacin (AK), CTX, ceftriaxone (CRO), CAZ, or aztreonam (ATM) and were incubated at 37 °C thereafter. The inhibited zone was measured using calipers according to CLSI 2019 guidelines. *K. pneumoniae* ATCC 700603 and *E. coli* ATCC 25922 were used as positive and negative control strains. Multidrug resistance (MDR) is defined as resistance to at least 1 agent in $\geq$3 antimicrobial classes.

## DNA extraction

DNA extraction was performed using AccuPrep Genomic DNA Extraction Kit (Bioneer, South Korea). Briefly, the bacterial suspension was centrifuged at 1,200 rpm, and the supernatant was removed. The pellet was washed with phosphate-buffered saline. After centrifugation, the supernatant was removed, and 20 μL of proteinase K was added. Tris-EDTA (TE) buffer was added to the DNA, and the final solution was adjusted for direct use as a polymerase chain reaction (PCR) template.
### Genotypic characterization of ESBL-producing isolates

Standard PCR was performed to screen the presence of seven ESBL-encoding genes: $bla_{TEM}$ (*Seenama, Thamlikitkul & Ratthawongjirakul, 2019*), $bla_{SHV}$ (*Seenama, Thamlikitkul & Ratthawongjirakul, 2019*), $bla_{CTX-M1}$ (*Le et al., 2015*), $bla_{CTX-M9}$ (*Le et al., 2015*), $bla_{GES}$ (*Bubpamala et al., 2018*), $bla_{VEB}$ (*Bubpamala et al., 2018*), and $bla_{PER}$ (*Bubpamala et al., 2018*) using specific primers described in Table 1. PCR reactions contained $1\times$ buffer, 1.5 mM $MgCl_2$, 400 µM dNTPs, 0.2 µM each of the forward and reverse primers, 1 U Taq polymerase, and 50 ng/µL DNA template. The PCR cycling conditions were dependent on the specific primers of the target gene as tabulated in Table 1. After PCR processing, PCR products were analyzed using agarose gel electrophoresis and sequenced on Applied Biosystems 3730XL (Macrogen, Seoul, Korea). The sequences were then aligned with those in the NCBI database using the BLAST search tool to determine sequence similarity (*Han et al., 2014*).

### Enterobacterial repetitive intergenic consensus-polymerase chain reaction (ERIC-PCR) and DNA amplification

The primers used for ERIC-PCR amplification were 5′-ATG TAA GCT CCT GGG GAT TCA C-3′ (F) and 5′-AAG TAA GTG ACT GGG GTG AGC G-3′ (R) (*Versalovic, Koeuth & Lupski, 1991*; *Fox et al., 2007*; *Ranjbar et al., 2017*). The total volume was 25 µL for each reaction, including 2 µL of template DNA (50 ng/ µL) (*E. coli* or *K. pneumoniae*), 5.2 µL of master mix, 0.25 µL of forward and reverse primers (100 µM), and 17.3 µL of deionized water. Subsequently, 30 cycles of denaturation (95 °C), annealing (48 °C), and extension (72 °C) for 60 s each were performed using a thermocycler. Deionized water and bacterial DNA of *E. coli* and *K. pneumoniae* strains were used as negative and positive controls, respectively.

### Polymerase chain reaction products and gel electrophoresis

Polymerase chain reaction products and agarose gel electrophoresis were conducted according to the methods of *Ranjbar et al. (2017)*, with modifications. The amplicon was electrophoresed on a 1.5% agarose gel in 0.5X TBE electrophoresis buffer. A 100-base pair (bp) DNA ladder was used as a standard marker. The gel bands were observed under ultraviolet light.

### Dendrogram and phylogenetic relationships

The DNA band pattern of the *E. coli* and *K. pneumoniae* samples obtained using gel electrophoresis of ERIC-PCR products was used as a generational structure for dendrogram analysis using BioNumerics version 7.1. For constructing a computerized dendrogram, the presence and absence of bands were presumed as 1 and 0, respectively. The dendrogram was then designed using the unweighted pair-group method with arithmetic mean, which is categorized in clustering methodologies and is based on clustering analysis.

Romyasamit et al. (2021), *PeerJ*, DOI 10.7717/peerj.11787
**Table 1 PCR primer designs and their conditions for gene amplifications.**

| Reference | Genes | Primer names | Primer sequences | Product size (bp) | Denature | | Annealing | | Extension | |
|---|---|---|---|---|---|---|---|---|---|---|
| | | | | | Temp (°C) | Time (s) | Temp (°C) | Time (s) | Temp (°C) | Time (s) |
| Le et al. (2015) | $bla_{CTX-M-1}$ | Ctxm1-115F | GAATTAGAGCGGCAGTCGGG | 588 | 95 | 30 | 60 | 90 | 72 | 90 |
| | | Ctxm1-702R | CACAACCCAGGAAGCAGGC | | | | | | | |
| | $bla_{CTX-M-9}$ | Ctxm9-16F | GTGCAACGGATGATGTTCGC | 475 | | | | | | |
| | | Ctxm9-490R | GAAACGTCTCATCGCCGATC | | | | | | | |
| Seenama, Thamlikitkul & Ratthawongjirakul (2019) | $bla_{TEM}$ | TEM-164.SE | TCGCCGCATACACTATTCTCAGAATGA | 445 | | | | | | |
| | | TEM-165.AS | ACGCTCACCGGCTCCAGATTTAT | | | | | | | |
| | $bla_{SHV}$ | SHV.SE | ATGCGTTATATTCGCCTGTG | 747 | 94 | 30 | 60 | 40 | 72 | 90 |
| | | SHV.AS | TGCTTTGTTATTCGGGCCAA | | | | | | | |
| Bubpamala et al. (2018) | $bla_{GES}$ | GES-F | TAC TGG CAG SGA TCG CTC AC | 838 | | | 62 | 90 | | |
| | | GES-F | TTG TCC GTG CTC AGG ATG AG | | | | | | | |
| | $bla_{VEB}$ | VEB-F | GCC AGA ATA GGA GTA GCA AT | 703 | 96 | 30 | 58 | 90 | 72 | 30 |
| | | VEB-R | TGG ACT CTG CAA CAA ATA CG | | | | | | | |
| | $bla_{PER}$ | PER-F | CTC AGC GCA ATC CCC ACT GT | 851 | | | 62 | 90 | | |
| | | PER-R | TTG GGC TTA GGG CAG AAA GCT | | | | | | | |

**Table 2** Summary of the raw vegetables and Prevalence of ESBL-producing *E. coli* and *K. pneumoniae* isolated from raw vegetables in retail markets in Mueang and Tha Sala districts, Nakhon Si Thammarat, Thailand.

| Order | Origin | *n* | No. of ESBL producer (%) | | |
|-------|--------|-----|------|------|-------|
| | | | EC | KP | Total |
| A | Thai yardlong beans | 34 | 1 (2.9) | 2 (5.9) | 3 (8.8) |
| B | Thai eggplant | 44 | 0 | 1 (2.3) | 1 (2.3) |
| C | Winged beans | 25 | 1 (4.0) | 1 (4.0) | 2 (8.0) |
| D | Young cashew leaves | 20 | 0 | 0 | 0 |
| E | Thai basil | 36 | 2 (5.6) | 1 (2.8) | 3 (8.3) |
| F | Cabbage | 21 | 0 | 0 | 0 |
| G | Cucumber | 36 | 1 (2.8) | 1 (2.8) | 2 (5.6) |
| H | Tomato | 31 | 0 | 2 (6.5) | 2 (6.5) |
| I | Long coriander | 32 | 0 | 1 (3.1) | 1 (3.1) |
| J | Lettuce | 26 | 0 | 0 | 0 |
| | Total | 305 | 5 (1.6%) | 9 (3.0%) | 14 (4.6) |

**Notes.**

Abbreviations: ESBL, extended-spectrum β-lactamase; EC, *E. coli*; KP, *K. pneumoniae*.

# RESULTS

## Prevalence of ESBL-producing *E. coli* and *K. pneumoniae*

ESBL-producing *E. coli* and *K. pneumoniae* isolates were found in 14 of the 305 samples obtained from seven out of 10 different vegetables (4.6% of the total) (Table 2). The highest frequencies of both ESBL-producing bacterial species were found in Thai yardlong beans (3/34, 8.8%) and Thai basil (3/36, 8.3%), followed by winged beans (2/25, 8%), tomato (2/31, 6.5%), and cucumber (2/36, 5.6%), with minimal detection in long coriander (1/32, 3.1%) and Thai eggplant (1/44, 2.3%). ESBL-producing isolates were not detected in young cashew leaves, cabbage, or lettuce (Tables 2 and 3).

ESBL-producing *K. pneumoniae* isolates were found more frequently than *E. coli* isolates at 9/14 (64.29%) and 5/14 (35.711%), respectively. Nine *K. pneumoniae* isolates that produced ESBL were variously distributed in several vegetables in this study. Five of the nine *K. pneumoniae* isolates (11.11%) were obtained from five different vegetables, *i.e.,* Thai eggplant, winged beans, Thai basil, cucumber, and long coriander, and four of the nine isolates (22%) were found in Thai yardlong beans and tomato (two each). Five ESBL-producing *E. coli* isolates were found in four different vegetables. Two isolates were most frequently found in Thai basil (40%), and one isolate (20%) was detected in each vegetable, including Thai yardlong beans, winged beans, and cucumber (Table 2).

Interestingly, both ESBL-producing *E. coli* and *K. pneumoniae* isolates were present in four specific vegetables, namely Thai yardlong beans, Thai basil, winged beans, and cucumber (Table 2). There was an equal proportion (1:1) of ESBL-producing *E. coli* and *K. pneumoniae* isolates in both winged beans and cucumber. Moreover, ESBL-producing *E. coli* and *K. pneumoniae* were frequently found in Thai basil and Thai yardlong beans. ESBL-producing *E. coli* isolates were detected more frequently in Thai basil than in Thai

**Table 3** Summary of the retail markets and Prevalence of ESBL-producing *E. coli* and *K. pneumoniae* isolated from raw vegetables in retail markets in Mueang and Tha Sala districts, Nakhon Si Thammarat, Thailand.

| Order | Origin | *n* | No. of ESBL producer (%) | | |
|---|---|---|---|---|---|
| | | | EC | KP | Total |
| 1 | Mor Chuan Fresh Market (Tha Sala M1) | 8 | 0 | 0 | 0 |
| 2 | Nab Anusorn Fresh Market (Tha Sala M2) | 38 | 1 (2.6) | 1 (2.6) | 2 (5.3) |
| 3 | Wat Node Market (Tha Sala M3) | 58 | 2 (3.5) | 1 (1.7) | 3 (5.2) |
| 4 | Thursday Flea Market (Tha Sala M4) | 53 | 0 | 0 | 0 |
| 5 | Sunday Flea market (Tha Sala M5) | 38 | 1 (2.6) | 0 | 1 (2.6) |
| 6 | Thapae Fresh Market (Mueang M1) | 42 | 1 (2.4) | 0 | 1 (2.4) |
| 7 | Khu Khwang Municipal Market (Mueang M2) | 21 | 0 | 4 (19.0) | 4 (19.0) |
| 8 | Tha Ma Fresh Market (Mueang M3) | 37 | 0 | 2 (5.4) | 2 (5.4) |
| 9 | Hua It Fresh Market (Mueang M4) | 10 | 0 | 1 (10.0) | 1 (10.0) |
| | Total | 305 | 5 (1.6%) | 9 (3.0%) | 14 (4.6) |

**Notes.**

Abbreviations: ESBL, extended-spectrum β-lactamase; EC, *E. coli*; KP, *K. pneumoniae*.

yardlong beans (2:1), which was in contrast to the proportion of ESBL-producing *K. pneumoniae* found in these vegetables (1:2) (Table 2).

Indeed, both ESBL-producing *E. coli* and *K. pneumoniae* were widely distributed in seven out of 10 edible vegetables and were more frequently found in Thai yardlong beans and Thai basil samples than in other vegetables. Individually, ESBL-producing *E. coli* was most frequently found in Thai basil, whereas ESBL-producing *K. pneumoniae* was most frequently found in Thai yardlong beans and tomato. Moreover, ESBL-producing *E. coli* were more frequently found in the Tha Sala district (4/5 isolates) than in the Mueang district (1/5 isolates).

## Antibacterial susceptibility phenotype

Eleven antibiotic agents of five classes (both β-lactam and non- β-lactam) were used to assess the antimicrobial susceptibility of ESBL-producing *E. coli* and *K. pneumoniae* isolates from 10 common edible vegetables in Southern Thailand using the disk diffusion method. All 14 isolates (100%) were strongly sensitive to IPM, followed by AK/CN, MEM/TE, ATM, CRO, CPD, CTX/CAZ, and AMP (Table 4).

All ESBL-producing *E. coli* isolates were sensitive to not only IPM but also CN with 100% susceptibility (5/5). The frequency of sensitivity in ESBL-producing *E. coli* isolates decreased in AK, TE/ATM, and AMP/MEM/CRO/CPD by 80% (4/5), 60% (3/5), and 40% (2/5), respectively. Furthermore, only one isolate (20%) exhibited intermediate activity against AK and CPD (Table 4). All ESBL-producing *K. pneumon* iae isolates were also determined to be susceptible to carbapenem IPM and MEM antibiotics (100%, 9/9). These strains were also highly sensitive to TE/AK (88.89%, 8/9), CN/ATM (77.78%, 7/9), CRO (66.67%, 6/9), and CTX/CAZ/CPD (55.56%, 5/9). Intermediate sensitivity was observed against AK and CRO (1/9 isolate of each) in *K. pneumoniae*, representing 11.11% of the total sample (Table 4).

Romyasamit et al. (2021), *PeerJ*, DOI 10.7717/peerj.11787

**Table 4 Antimicrobial susceptibility test in ESBL-producing *E.coli* and *K.pneumoniae*.**

| Mode of action | Drug classes | Drug names (Abbreviation) | EC and KP (*n* = 14) | | | EC (*n* = 5) | | | KP (*n* = 9) | | |
|---|---|---|---|---|---|---|---|---|---|---|---|
| | | | R | S | I | R | S | I | R | S | I |
| Targeted β-lactam ring | Aminopenicillin | Ampicillin (AMP) | 12 (85.71) | 2 (14.29) | 0 | 3 (60) | 2 (40) | 0 | 9 (100) | 0 | 0 |
| | Carbapenem | Imipenem (IPM) | 0 | 14 (100) | 0 | 0 | 5 (100) | 0 | 0 | 9 (100) | 0 |
| | | Meropenem (MEM) | 3 (21.43) | 11 (78.57) | 0 | 3 (60) | 2 (40) | 0 | 0 | 9 (100) | 0 |
| | | Aztreonam (ATM) | 4 (28.57) | 10 (71.43) | 0 | 2 (40) | 3 (60) | 0 | 2 (22.22) | 7 (77.78) | 0 |
| | Cephalosporin | Cefotaxime (CTX) | 9 (64.29) | 5 (35.71) | 0 | 5 (100) | 0 | 0 | 4 (44.44) | 5 (55.56) | 0 |
| | | Ceftriaxone (CRO) | 5 (35.71) | 8 (57.14) | 1 (7.14) | 3 (60) | 2 (40) | 0 | 2 (22.22) | 6 (66.67) | 1 (11.11) |
| | | Ceftazidime (CAZ) | 9 (64.29) | 5 (35.71) | 0 | 5 (100) | 0 | 0 | 4 (44.44) | 5 (55.56) | 0 |
| | | Cefpodoxime (CPD) | 6 (42.86) | 7 (50.00) | 1 (7.14) | 2 (40) | 2 (40) | 1 (20) | 4 (44.44) | 5 (55.56) | 0 |
| Non targeted β-lactam ring | Aminoglycoside | Gentamycin (CN) | 2 (14.29) | 12 (85.71) | 0 | 0 | 5 (100) | 0 | 2 (22.22) | 7 (77.78) | 0 |
| | | Amikacin (AK) | 0 | 12 (85.71) | 2 (14.29) | 0 | 4 (80) | 1 (20) | 0 | 8 (88.89) | 1 (11.11) |
| | Tetracycline | Tetracycline (TE) | 3 (21.43) | 11 (78.57) | 0 | 2 (40) | 3 (60) | 0 | 1 (11.11) | 8 (88.89) | 0 |

The *Antibiotics* columns (Mode of action, Drug classes, Drug names (Abbreviation)) fall under the heading **Antibiotics**, and the R/S/I grouped columns fall under **No. of ESBL-producing isolates (%)**.

**Notes.**

Abbreviations: ESBL, extended-spectrum β-lactamase; EC, *E. coli*; KP, *K. pneumoniae*; R, resistance; S, Susceptible; I, intermediate.

The presence of IPM and AK sensitivity against ESBL-producing *E. coli* and *K. pneumoniae* is promising; however, these strains were also resistant to several antibiotics (Table 4 and 5). In total ($n = 14$), both ESBL producers (12) were found to be resistant to AMP (85.71%), followed by CTX/CAZ (9, 64.29%), CPD (6, 42.86%), CRO (5, 35.71%), ATM (4, 28.57%), MEM/TE (3, 21.43%), and CN (2, 14.29%).

The antibiotic-resistance profile of ESBL-producing *E. coli* ($n = 5$) was 100% for CTX/CAZ (5/5), 60% for AMP/MEM/CRO (3/5), and 40% for TE/CPD/ATM (2/5). All isolates of ESBL-producing *K. pneumoniae* ($n = 9$) were completely resistant to AMP (100%), followed by CTX/CAZ/CPO (4, 44.44%), CN/CRO/ATM (2, 22.22%), and TE (1, 11.11%) (Table 4).

## Multidrug resistance patterns

MDR is defined as bacterial resistance to antibiotics or to at least one agent in $\geq 3$ antimicrobial classes (*Magiorakos et al., 2012*). Here 7 out of 14 (50%) *E. coli* and *K. pneumoniae* isolates were MDR. Of these, three were *E. coli* MDR isolates (60%) and four were *K. pneumoniae* MDR isolates (44.44%) (Tables 5 and 6). Interestingly, the *E. coli* isolate (A60301) obtained from Thai yardlong beans at a market in Mueang exhibited more resistance (63.64%) than sensitivity (36.36%). MDR analysis revealed that this isolate was resistant to four classes of antibiotics and was resistant to 7 out of 11 antibiotics in total. Another isolate (E50501) of ESBL-producing *E. coli* from Thai basil in Tha Sala also had an MDR profile including four antibiotic classes and was resistant to 5 out of 11 antibiotics. However, this isolate had slightly more antibiotic sensitivity (54.55%) than resistance (45.45%). The proportional sensitivity was similar to that of an MDR isolate (C30501). An MDR *E. coli* isolate derived from an MDR *K. pneumoniae* (A80301) was also isolated from Thai yardlong beans from Mueang district and was resistant to four classes of antibiotics (6/11) and exhibited greater resistance (54.55%) than sensitivity (45.45%). Although the *K. pneumoniae* isolates (B90101, C70101, and G70301) were obtained from several vegetables (Thai eggplant, winged bean, and cucumber, respectively) in different Mueang markets, all exhibited MDR to three classes of antibiotics (3/11), though these isolates were more susceptible (72.73%) to eight other antibiotics than resistant (27.27%). Consequently, these MDR patterns were more frequently found in *E. coli* than in *K. pneumoniae* isolates and were more frequently detected in samples from Mueang markets, which had the highest proportion of contamination in Thai yardlong beans (Table 6). Several of the above isolates showed MDR activity, but their sensitivities to other antibiotics ensure that the infection caused by them is still treatable.

We then characterized seven ESBL-encoding gene variants ($bla_{TEM}$, $bla_{SHV}$, $bla_{CTX-M1}$, $bla_{CTX-M9}$, $bla_{GES}$, $bla_{VEB}$, and $bla_{PER}$) by PCR and confirmed their identity by DNA sequencing. This demonstrated that the $bla_{SHV}$-ESBL variant (57.14%) (8/14) was the most prevalent ESBL-encoding variant in *E. coli* and *K. pneumoniae* (Table 7), with four out of 14 (28.57%) isolates from each species containing this variant. Notably, none of the other ESBL-coding variants were detected in *K. pneumoniae* isolates. Furthermore, MDR genes were detected in only few of the *E. coli* isolates, with two (40%) and one (20%) being

**Table 5  Antimicrobial susceptibility test against ESBL-producing *E. coli* and *K. pneumoniae* isolates.**

| ESBL-producing isolates | Antibiotics classes and antibiotics | | | | | | | | | | |
|---|---|---|---|---|---|---|---|---|---|---|---|
| | Targeted β-lactam ring | | | | | | | | Non-targeted β-lactam ring | | |
| | Aminopenicillin | Carbapenem | | | Cephalosporin | | | | Aminoglycoside | | Tetracycline |
| | Ampicillin (AMP) | Imipenem (IPM) | Meropenem (MEM) | Aztreonam (ATM) | Cefotaxime (CTX) | Ceftriaxone (CRO) | Ceftazidime (CAZ) | Cefpodoxime (CPD) | Gentamicin (CN) | Amikacin (AK) | Tetracycline (TE) |
| EC | | | | | | | | | | | |
| A60301 | R | S | R | S | R | R | R | R | S | S | R |
| C30501 | R | S | R | R | R | S | R | S | S | S | S |
| E20301 | S | S | S | R | R | R | R | R | S | S | S |
| E50501 | R | S | R | S | R | S | R | S | S | S | R |
| G30501 | S | S | S | S | R | R | R | I | S | I | S |
| KP | | | | | | | | | | | |
| A70201 | R | S | S | S | S | S | R | S | S | I | S |
| A80301 | R | S | S | S | R | R | R | R | R | S | S |
| B90101 | R | S | S | S | S | S | S | R | S | S | R |
| C70101 | R | S | S | R | S | S | S | S | R | S | S |
| E20502 | R | S | S | S | R | I | S | R | S | S | S |
| G70301 | R | S | S | R | R | S | S | S | S | S | S |
| H30301 | R | S | S | S | S | S | R | R | S | S | S |
| H80301 | R | S | S | S | S | S | R | S | S | S | S |
| I70201 | R | S | S | S | R | R | S | S | S | S | S |

**Notes.**

Abbreviations: ESBL, extended-spectrum β-lactamase; EC, *E. coli*; KP, *K. pneumoniae*; R, resistance; S, Susceptible; I, intermediate.

positive for $bla_{TEM}$ and $bla_{CTX-M1}$, $bla_{CTX-M9}$, and $bla_{GES}$ variants, respectively. Moreover, $bla_{VEB}$ or $bla_{PER}$ were not detected in any of the *E. coli* or *K. pneumoniae* isolates.

Multiple ESBL-encoding genes were found in the *E. coli* isolates (2/5). A60301 and C30501 were isolated from Thai yardlong beans and winged beans from Mueang and Tha Sala markets, respectively, including two sets of three ESBL-coding gene variants: $bla_{TEM}$, $bla_{CTX-M1}$, $bla_{GES}$ and $bla_{TEM}$, $bla_{SHV}$, $bla_{CTX-M9}$. Each ESBL gene variant amplified from *E. coli* isolates was detected by agarose gel electrophoresis (Fig. 1).

## Dendrogram relationship of isolated strains of *E. coli* and *K. pneumoniae*

Phylogenetic trees of five *E. coli* isolates were generated by GelClust (Fig. 2A). This demonstrated that the A60301 isolate was more closely related to C30501 than to other *E. coli* isolates, with an identity of approximately 73%. The G30501 isolate was probably another strain related to the E20301 and E50501 isolates (with 77% identity). The E20301 and E50501 strains were more closely related to each other than to other isolates (~84% identity). Phylogenetic analysis of the five *E. coli* isolates could thus be used for individual strain identification. Each isolate of *E. coli* obtained from the same vegetable or market was a different strain, but as E20301 and E50501 came from the same vegetables, they were likely to be closely related.

We also created a phylogenetic tree for all isolated *K. pneumoniae* strains, which had a wider range of genetic heterogeneities than *E. coli* isolates (Fig. 2B). The cluster analysis

**Table 6  Multidrug resistance patterns of ESBL-producing *E. coli* and *K. pneumoniae* isolates.**

| ESBL-producing isolates | No. of antibiotics activity (%) | | | MDR patterns | No. of resistant antibiotics classes | MDR activity |
|---|---|---|---|---|---|---|
| | Antibiotics (*n* = 11) | | | | | |
| | R | S | I | | | |
| | | | | EC (*n* = 5) | | |
| A60301 | 7 (63.64) | 4 (36.36) | 0 | AMP, MEM, CTX, CRO, CAZ, CPD, TE | 4 | + |
| C30501 | 5 (45.45) | 6 (54.55) | 0 | AMP, MEM, ATM, CTX, CAZ | 3 | + |
| E20301 | 5 (45.45) | 6 (54.55) | 0 | ATM, CTX, CRO, CAZ, CPD | 2 | − |
| E50501 | 5 (45.45) | 6 (54.55) | 0 | AMP, MEM, CTX, CAZ, TE | 4 | + |
| G30501 | 3 (27.27) | 6 (54.55) | 2 (18.18) | CTX, CRO, CAZ | 1 | − |
| | | | | KP (*n* = 9) | | |
| A70201 | 2 (18.18) | 8 (72.73) | 1 (9.09) | AMP, CAZ | 2 | − |
| A80301 | 6 (54.55) | 5 (45.45) | 0 | AMP, CN, CTX, CRO, CAZ, CPD | 3 | + |
| B90101 | 3 (27.27) | 8 (72.73) | 0 | AMP, CPD, TE | 3 | + |
| C70101 | 3 (27.27) | 8 (72.73) | 0 | AMP, CN, ATM, | 3 | + |
| E20502 | 3 (27.27) | 7 (63.64) | 1 (9.09) | AMP, CTX, CPD | 2 | − |
| G70301 | 3 (27.27) | 8 (72.73) | 0 | AMP, ATM, CTX | 3 | + |
| H30301 | 3 (27.27) | 8 (72.73) | 0 | AMP, CAZ, CPD | 2 | − |
| H80301 | 2 (18.18) | 9 (81.82) | 0 | AMP, CAZ | 2 | − |
| I70201 | 3 (27.27) | 8 (72.73) | 0 | AMP, CTX, CRO | 2 | − |

Notes.

Abbreviations: ESBL, extended-spectrum β-lactamase; EC, *E. coli*; KP, *K. pneumoniae*; R, resistance; S, susceptible; I, intermediate; MDR, multidrug resistance; AMP, ampicillin; IPM, imipenem; MEM, meropenem; ATM, aztreonam; CTX, cefotaxime; CRO, ceftriaxone; CAZ, ceftazidime; CPD, cefpodoxime; CN, gentamicin; AK, amikacin; TE, tetracycline.

and related dendrogram revealed that five molecular genetic clusters were represented individually in each strain of *K. pneumoniae*. The H80301 strain was similar to I70201 (85% identity), although these were the least related to any of the other stains (Fig. 2B). G70301 was related with 67% identity to A80301/E20502 and had 55% identity with H30301. The A80301 and E20502 isolates were shown to be of the same strain. Moreover, H30301 showed 60% identity with C70101. The final clusters delineated the relationship between C70101 and A70201/B90101 isolates, which had 58% identity. A70201 and B90101 were of different strains with 67% identity. The *K. pneumoniae* strains showed high diversity with respect to the vegetables and markets, with each isolate coming from an independent source.

# DISCUSSION

This is the first report describing the number of ESBL producers contaminating Thai basil. In this study, we determined that the highest prevalence of isolates of ESBL-producing *E. coli* and *K. pneumoniae* producers is in Thai yardlong beans and Thai basil samples, representing 8.3%–8.8% of all samples. Nevertheless, the prevalence of ESBL-producing *E. coli* and *K. pneumoniae* was lower when compared with that of other studies (different

**Table 7  ESBL-encoding genes of *E. coli* and *K. pneumoniae* isolates.**

| ESBL-producing isolates | ESBL-coding genes (%) | | | | | | |
|---|---|---|---|---|---|---|---|
| | $bla_{TEM}$ | $bla_{SHV}$ | $bla_{CTX-M1}$ | $bla_{CTX-M9}$ | $bla_{GES}$ | $bla_{VEB}$ | $bla_{PER}$ |
| EC (*n* = 5) | | | | | | | |
| A60301 | + | – | + | -. | + | – | – |
| C30501 | + | + | – | + | – | – | – |
| E20301 | – | + | – | – | – | – | – |
| E50501 | – | + | – | – | – | – | – |
| G30501 | – | + | – | – | – | – | – |
| Total EC | 2 (40) | 4 (80) | | 1 (20) | | 0 | 0 |
| KP (*n* = 9) | | | | | | | |
| A70201 | – | + | – | – | – | – | – |
| A80301 | – | – | – | – | – | – | – |
| B90101 | – | + | – | – | – | – | – |
| C70101 | – | – | – | – | – | – | – |
| E20502 | – | – | – | – | – | – | – |
| G70301 | – | + | – | – | – | – | – |
| H30301 | – | – | – | – | – | – | – |
| H80301 | – | + | – | – | – | – | – |
| I70201 | – | – | – | – | – | – | – |
| Total KP | 0 | 4 (44.44) | 0 | 0 | 0 | 0 | 0 |
| Total EC and KP (*n* = 14) | 2 (14.29) | 8 (57.14) | | 1 (7.14) | | 0 | 0 |

**Notes.**

Abbreviations: ESBL, extended-spectrum β-lactamase; EC, *E. coli*; KP, *K. pneumoniae*; +, positive; -, negative.

types and sample sizes of vegetables), with frequencies of 10.1%, 13.3%, 16.4%, and 43.3% (*Kim et al., 2015*; *Richter et al., 2019*; *Ye et al., 2018*; *Zurfluh et al., 2015*). Therefore, this may imply that the variation in the prevalence of ESBL-producing strains depends on the sample size and types of vegetables tested.

We determined that the *E. coli* and *K. pneumoniae* strains isolated in this study showed resistance to a range of antibiotics. These isolates were resistant to broad-spectrum AMP (85.71%) and third-generation cephalosporin antibiotics, including CTX, CRO, CAZ, and CPD (40%–70%). Our results are in accordance with those of a previous review by *Falagas & Karageorgopoulos (2009)* that showed that *E. coli* and *K. pneumoniae* produced high amounts of ESBL enzyme, which can hydrolyze many common antibiotics, *i.e.,* penicillin (antibiotic class same as AMP), cephalosporins, and ATM (carbapenem), resulting in antibiotic resistance in these bacteria. Furthermore, *Zurfluh et al., (2015)* reported that 26 isolates of ESBL-producing *E. coli* and *K. pneumoniae* obtained from imported vegetables in Dominican Republic, India, Thailand, and Vietnam were resistant to AMP (100%) and narrow-spectrum cephalosporins: cephalothin (INN) and CTX (88.3%). *Kim et al. (2015)* determined the antibiotic susceptibility of ESBL-producing *E. coli* and *K. pneumoniae*, isolated from sprout samples in South Korea, and showed these were resistant to CTX (100%). Another study showed that all ESBL-producing *E. coli* isolated from raw vegetables were completely resistant to AMP, CTX, and other antibiotics (piperacillin, cefazoline, and

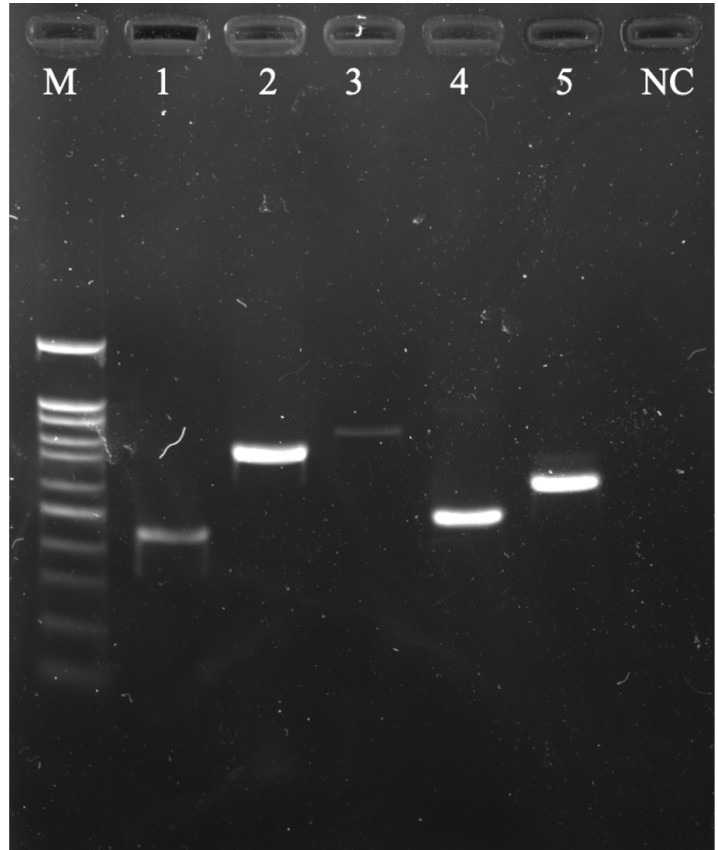

**Figure 1** **A 1.5% (w/v) agarose gel image showing the ESBL-encoding genes amplified from *E. coli* isolates isolated from vegetables using PCR method.** Lane *M* = 100 bp DNA Ladder, Lane 1 = *TEM* gene fragments amplified from *E. coli* A60301 isolates, Lane 2 = *SHV* gene fragments amplified from *E. coli* G3051 isolates, Lane 3 = *GES* gene fragments amplified from *E. coli* A6031 isolates, Lane 4 = *CTX-M9* gene fragments amplified from *E. coli* C30501 isolates, Lane 5 = *CTX-M1* gene fragments amplified from *E. coli* A60301 isolates, and Lane NC = Negative control.

nalidixic acid) (*Song et al., 2020*). *E. coli* isolated from fresh vegetables showed resistance to cephalosporins, such CTX and CAZ (92%) (*Ye et al., 2018*). In our study, *E. coli* was most frequently resistant to cephalosporins, such as CTX and CAZ (100%), and AMP (60%), which was not only consistent with the above reports for isolates from raw vegetables but also for isolates from clinical specimens (60%–62% to cephalosporins) and healthy subjects (72% to AMP) in low- and middle-income countries (*Nji et al., 2021*; *Teklu et al., 2019*; *Ye et al., 2018*; *Song et al., 2020*). The resistance patterns of *E. coli* from different sources may be associated with each other because of transmission of resistance genes between humans and environments. Moreover, in this study, MDR patterns of *E. coli* (21.43%) and *K. pneumoniae* (28.57%) were present in 50% of ESBL-producing isolates that shared resistance to 3–4 antibiotic classes of β-lactam antibiotics (aminopenicillin, cephalosporin, and carbapenem) and non- β-lactam antibiotics (aminoglycosides and TE). Our results correlated with those of a previously study that described MDR strains, particularly against β-lactam CTX and non- β-lactam CHL and TE, found in iceberg
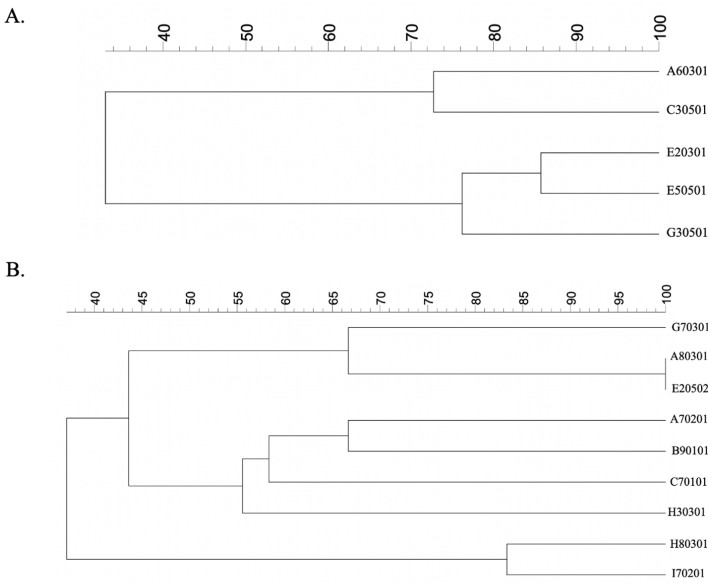

**Figure 2** ERIC-PCR profiles of (A) ESBL-producing *E. coli* isolates and (B) ESBL-producing *K. pneumoniae* isolates from raw vegetables. A dendrogram was generated from the ERIC-PCR typing of five ESBL-producing *E. coli* and nine *K. pneumoniae* isolates. The analysis was performed using BioNumerics fingerprint data software v7.1 and the unweighted pair group method with arithmetic means (UPGMA) clustering method and the dice similarity coefficient with 1% optimization and 1.5% position tolerance. EC, *E. coli*; KP, *K. pneumoniae*.

lettuce (*Bhutani et al., 2015*). An MDR *E. coli* strain that was resistant to β-lactam (AMP, INN, and CTX) and non- β-lactam (trimethoprim and TE) antibiotics was isolated from imported Thai yardlong beans in Switzerland (*Zurfluh et al., 2015*). Two MDR *E. coli* strains isolated from sprouts were resistant to CTX, CAZ, cefepime, ATM, ciprofloxacin (CIP), and trimethoprim/sulfamethoxazole (*Kim et al., 2015*). We also found strong resistance to multiple antibiotic classes in *E. coli*, highlighting concerns of MDR. Although the presence of MDR in *K. pneumoniae* isolates was higher than that in *E. coli* isolates, other antibiotic sensitivities remained high (except for AMP). Conversely, *Bhutani et al. (2015)* reported major resistance to CTX (86%) in two *K. pneumoniae* isolates from iceberg lettuce.

The *K. pneumoniae* strains isolated here commonly carried the $bla_{SHV}$ variant, which was also present in several *E. coli* strains. However, *E. coli* isolates from selected vegetables seem be strong producers of ESBL enzymes *via* the co-existence of several ESBL-encoding genes. The *E. coli* A60301 and C30501 strains isolated from yardlong beans and winged beans carried the $bla_{TEM}$, $bla_{CTX-M1}$, and $bla_{GES}$ and the $bla_{TEM}$, $bla_{SHV}$, and $bla_{CTX-M9}$ variants, respectively. Their co-expression may result in stronger ESBL production and increased antibiotic resistance or MDR. Consequently, *E. coli* strains producing high levels of ESBL may be harbored in Thai yardlong beans and wing beans. However, the level of ESBL production is yet to be evaluated.

Not surprisingly, the $bla_{SHV}$ variant was frequently found both in *E. coli* and *K. pneumoniae* because this variant is a common genetic variant found in the ESBL-producing

Enterobacteriaceae family (*Pitout & Laupl, 2008*; *Zurfluh et al., 2015*). $bla_{SHV}$ and $bla_{TEM}$ were recognized as producing ESBL and were initially reported from clinical isolates, as were the $bla_{VEB}$, $bla_{PER}$, and $bla_{GES}$ variants, although five types of $bla_{CTX-M}$ were originally mainly found in environmental isolates (*Kluyvera* spp.) (*Pitout & Laupl, 2008*; *Zurfluh et al., 2015*). The relevant genes are thought to have been mobilized into conjugative plasmids, and thus transferred to pathogenic bacteria (*Zurfluh et al., 2015*). Community-onset ESBL-associated infections, including urinary infection and sepsis are principally caused by *E. coli* having $bla_{CTX-M}$. ESBLs (*Zurfluh et al., 2015*). Therefore, the relevant genes might contribute to the presence of ESBLs in community-transmitted infections from animal sources to humans *via* food chains or patient-to-patient transmission (*Zurfluh et al., 2015*). Notably, in this study, the A60301 and C30501 *E. coli* isolates that co-harbored those relevant ESBL variants may be considered as antibiotic-resistance predictors that can pass between food chains (vegetable) and undergo human transmission in the community if consumers lack good hygiene.

Carbapenems are widely regarded as the antibiotics of choice for the treatment of severe infections caused by ESBL-producing Enterobacteriaceae (*Pitout & Laupl, 2008*). In our study, the antibiotic susceptibility of ESBL-producing *E. coli* and *K. pneumoniae* isolates was 100% (completely resistant) to carbapenem (IPM) and 85.71% (mostly resistant) to aminoglycosides such as AK and CN. Carbapenem sensitivities similar to these have also been reported in other studies (83% and 99.3%–100%) (*Ye et al., 2018*; *Devrim et al., 2011*).

The prevalence of *E. coli* and *K. pneumoniae* isolates was demonstrated by the range of different strains determined using phylogenetic tree analysis. It was clearly shown that ESBL-producing *E. coli* and *K. pneumoniae* outbreaks have been problem in the food chain. However, further analysis using whole-genome sequence is essential to determine more epidemiological features of the ESBL-producing *E. coli* and *K. pneumoniae* in raw vegetables.

## CONCLUSIONS

As per our findings, it was determined that retail vegetables at local markets in Nakhon Si Thammarat may be reservoirs for the spread of ESBL-producing *E. coli* and *K. pneumoniae*. This may lead to antibiotic resistance through food chain transmission if consumer hygiene is poor. Nevertheless, further investigations are necessary to understand supplementary epidemiological features of ESBL-producing *E. coli* and *K. pneumoniae* in raw vegetables.

## ACKNOWLEDGEMENTS

The authors thank the Research Institute for Health Sciences Walailak University, School of Allied Health Sciences, Walailak University, and Department of Biomedical Sciences, Faculty of Medicine, Prince of Songkla University for providing the required laboratory instruments.

### Funding

This work was supported by the Research Institute for Health Sciences, Walailak University [grant number; WU-IRG-63-085]. The funders had no role in study design, data collection and analysis, decision to publish, or preparation of the manuscript.

### Grant Disclosures

The following grant information was disclosed by the authors:
Research Institute for Health Sciences, Walailak University: WU-IRG-63-085.

### Competing Interests

The authors declare there are no competing interests.

### Author Contributions

- Chonticha Romyasamit conceived and designed the experiments, performed the experiments, analyzed the data, prepared figures and/or tables, authored or reviewed drafts of the paper, and approved the final draft.
- Phoomjai Sornsenee conceived and designed the experiments, analyzed the data, prepared figures and/or tables, and approved the final draft.
- Siriphorn Chimplee analyzed the data, authored or reviewed drafts of the paper, and approved the final draft.
- Sitanun Yuwalaksanakun performed the experiments, authored or reviewed drafts of the paper, and approved the final draft.
- Dechawat Wongprot and Phanvasri Saengsuwan performed the experiments, prepared figures and/or tables, and approved the final draft.

### Data Availability

The nucleotide sequences are available at NCBI GenBank: blaTEM, MW822683.1; blaCTX-M1, MW822679; blaGES, MW822682; blaSHV, MW822681; and blaCTX-M9, MW822680.

### Supplemental Information

Supplemental information for this article can be found online at http://dx.doi.org/10.7717/peerj.11787#supplemental-information.

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
