# Peer review of "Prevalence and characterization of extended-spectrum β-lactamase-producing Escherichia coli and Klebsiella pneumoniae isolated from raw vegetables retailed in Southern Thailand"

_PeerJ, doi:10.7717/peerj.11787_

## Round 0.1 · original submission · Major Revisions

Please address the concerns of all reviewers and amend your manuscript accordingly.

·

Basic reporting

Language:
The English language was clear. Some suggested changes were made in the manuscript.
Literature references, sufficient field background/context provided:
Abstract, results section: In line 32 of the abstract additional information is needed as indicated in the comments box in the manuscript: The title of the manuscript mentioned prevalence of ESBL producing Escherichia coli and Klebsiella pneumoniae. In the first section of the results the following statement was made: ESBL-producing E. coli and K. pneumoniae isolates were found in 14 of the 305 samples obtained from seven out of ten types of vegetables (4.6% of the total). It is important to incorporate information on prevalence in the vegetables, before describing the genes screened for in the abstract.

Introduction:
• In the abstract broad-spectrum ampicillin and extended-spectrum cephalosporin resistance of clinical isolates of E. coli and K. pneumoniae were mentioned specifically. The author tested isolate resistance to ampicillin which is a broad-spectrum antibiotic, cephalosporins, carbapenems, tetracycline and aminoglycosides. Some background information on the importance of ampicillin resistance of E. coli and K. pneumoniae should be included in the introduction.

• Some background information regarding the specific genes screened for is lacking in the introduction.

• In lines 54 and 57 references should be added as indicated with track changes in the text in the introduction.
• Line 57-58 some detail of the organisms i.e. E. coli and K. pneumoniae should be mentioned and not only generally referring to microbes to clarify the focus and provide more context for performing the study.

Professional article structure, figures, tables:
Table 3: The origin of the samples not written in the English alphabet. Please correct.
Table 3: The heading mentions Eric-PCR primers, however these were included in the manuscript text and are not listed in the table.

Self contained with relevant results to hypothesis:
The manuscript form a distinct unit of research question to be addressed, results obtained and discussed.

Experimental design

Original primary research within the Aims and Scope of the journal:
The manuscript focuses on isolation, identification and characterisation of potential human pathogenic bacteria, specifically ESBL producing E. coli and K. pneumoniae and therefore fits into Biological/Environmental and potentially Health Sciences which are in the scope of the journal.
Research question well defined, relevant & meaningful. It I stated how research fills and identified knowledge gap.
The motivation for the study, goals, the knowledge gap to be filled and the contribution to existing knowledge were clearly articulated.
Rigorous investigation performed to a high technical & ethical standard.
Some suggestions were made regarding the methodology used in this study.
Methods described with sufficient detail & information to replicate.
• In line 78 please change the heading Bacteria identification process and ESBL-producing bacteria isolation to reflect the specific organisms targeted.

• In line 84 please include information how the colonies were purified.

• In line 98 add the reference details.

• In line 11 add the original reference for the Kirby-Bauer test.

• In line 118 add the reference.

• In line 124: Title changed. See track changes in the manuscript.

• In line129 please describe how the concentration of the DNA was determined and what was the range of concentrations used for PCR analysis. Also how was the DNA template concentration dependant on the primers used?

• In line 130 and in Table 1 please add reference/s for the primers used in PCR analysis to screen for and identify the ESBL encoding genes.

Validity of the findings

In Table 2 the heading states common edible vegetables from a market. This is confusing as in the Material and Methods two markets were described.

Is it not a summary of the vegetables and ESBL prevalence in both markets as reflected in Table 3? If that is the case the heading has to be corrected.

As indicated previously, please write origin details in Table 3 using the English alphabet.

Please refer to the track changes in the results and discussion section of the manuscript as there are a number of questions/changes.

Additional comments

It is highly recommended that the authors determine the variants of the amplicons sequenced to confirm and discuss the identities of the genes detected which will add significant value to the results obtained during the study and comparison to results reported in other studies. It certainly is a worthwhile study to report, but after substantial revision.

This can be performed by subjecting amplicon sequences of the ESBL genes to BLAST nucleotide search analysis to identify the antimicrobial resistance genes and variants.

·

Basic reporting

English - it is clear

Literature - ok

Article structure - ok

Hypothesis - well-stated and relevant

Experimental design

Experimental design - ok for the study

Research question - clearly defined and important

Methodology - for what was done it is ok

Validity of the findings

The findings are valid.

My only suggestion is that what is missing is whole-genome sequencing (WGS). It would be important to understand the level of genomic variation, genotypes, and global picture of AMR and other putative properties based on pan-genomic analysis. I would even suggest that the population structure analysis of the isolates (using genomic data) be done in the context of freely available genomes of the same species deposited on NCBI, by specifically picking representative genomes from ST (sequence types) from different environments. This is just a suggestion. Also, I would use WGS with more than just one isolate from each plate and try to pick more colonies to maximize the chances of capturing different variants. I think it is important to get WGS data to link genotypes to phenotypes at a higher level of genotypic resolution than what is provided now.

Additional comments

I think the work is important and has implications for food safety. But in my view, it can maximize its reach by using WGS.

Reviewer 3 ·

Basic reporting

The manuscript is clear and easy to follow. Methods and Results are described thoroughly, although there are some places where additional might be useful.

Experimental design

The experimental approaches are well defined.

Validity of the findings

The manuscript described a meaningful assessment of the ESBL-producing E.coli and K.pseumonae presence in raw vegetables in Thailand, which is of significant value to public health and food safety. All data are presented in a clear manner, and well supported the conclusions.

Additional comments

The manuscript described a thorough characterization of extended spectrum beta-lactamase producing Klebsiella and E.coli in raw vegetables in local markets in Thailand. The research presented in this manuscript is scientifically sound, and the conclusions are well supported by the data. There are, however, some minor issues that I hope the authors can address before this manuscript can be accepted for publication in PeerJ.

Line 122, 333, 337, 339 etc.: please make sure gene names are italicized throughout the manuscript

It would be useful to include in the introduction the structures of the antibiotic classes used in this study, and the phylogenetic relationship of the b-lactamase genes characterized, as well as their targets on the b-lactam structures.


Figure 1:
I suggest the authors to include more description of the experiment in the figure legends.

Figure 2: It is important to clarify how the dendrogram was constructed (from ERIC-PCR patterns), and how ERIC-PCR pattern is representative of the phylogenetic relationship between the isolates. Therefore, I would like to state again that the authors should clarify the relationship between the b-lactamase genes in this study.

Does the presence of multiple drug resistance in these isolates suggest anything about the possibility of horizontal gene transfer among them, given that they are isolated from the very same location/sample? Can the authors elaborate more on this?

E. coli and Klebsiella are human pathogens and not natural plant colonizers. Can the authors discuss a source of contamination on the raw vegetables collected from the local market?

Table 4: the title should be corrected as “Antimicrobial susceptibility test in E.coli and K.pneumoniae”, as prior this test, it was unknown whether these isolates are resistant to an extended spectrum of b-lactam.

---

## Round 0.2 · Major Revisions

Please address the remaining critiques of the reviewer and amend your manuscript accordingly

·

Basic reporting

The language and structure of the article are clear after revisions were made.

Experimental design

I have no comments here.

Validity of the findings

I understand the funding argument posed to guard against doing WGS for this study. I empathize with the author in all honesty. I truly do.

But after doing such a nice study in attempting to isolate important AMR bacteria that has consequences for Public Health, the work necessitates WGS.

If you examine the discussion, the authors would like to compare what you have in light of the population structure of these isolates and those associated with humans or other sources. In my view, that cannot be done with the data you have.

Examine these two paragraphs:

"The prevalence of E. coli and K. pneumoniae isolates was demonstrated by the range of different strains determined using phylogenetic tree analysis. ESBL-producing bacteria may, therefore, be involved in spread to community, thereby causing infectious diseases.

Conclusions
As per our findings, it was determined that retail vegetables at local markets in Nakhon Si Thammarat may be reservoirs for the spread of ESBL-producing E. coli and K. pneumoniae. This may lead to antibiotic resistance through food chain transmission if consumer hygiene is poor."

If you had WGS and analyzed these data in the context of other genomes deposited on NCBI coming from humans and other sources, these statements would carry enough weight to provide a hypothesis or even a prediction. Because I believe this work is relevant for Public Health, WGS is crucial here.

Additional comments

I respect the work done by the authors. I think this work has the potential to be even stronger by WGS of the isolates acquired in the study, and a population-based analysis in light of other clinical and environmental isolates.

---

## Round 0.3 · accepted · Accept

Thank you for providing an honest response to the reviewer's suggestions. I decided that the data you provided in this manuscript are important and sufficient for publication. Therefore, I am accepting the manuscript in its present form.